# South American National Contributions to Knowledge of the Effects of Endocrine Disrupting Chemicals in Wild Animals: Current and Future Directions

**DOI:** 10.3390/toxics10120735

**Published:** 2022-11-28

**Authors:** Sylvia Rojas-Hucks, Ignacio A. Rodriguez-Jorquera, Jorge Nimpstch, Paulina Bahamonde, Julio A. Benavides, Gustavo Chiang, José Pulgar, Cristóbal J. Galbán-Malagón

**Affiliations:** 1Departamento de Ecología y Biodiversidad, Facultad Ciencias de la Vida, Universidad Andres Bello, República 440, Santiago 8370134, Chile; 2Centro de Humedales Río Cruces (CEHUM), Universidad Austral de Chile, Valdivia 5090000, Chile; 3Facultad de Ciencias, Instituto de Ciencias Marinas y Limnológicas, Universidad Austral de Chile, Valdivia 5090000, Chile; 4Laboratory of Aquatic Environmental Research, Centro de Estudios Avanzados—HUB Ambiental UPLA, Universidad de Playa Ancha, Valparaíso 2360004, Chile; 5Millennium Nucleus of Austral Invasive Salmonids (INVASAL), Concepción 4070386, Chile; 6Cape Horn International Center (CHIC), Universidad de Magallanes, Punta Arenas 6210427, Chile; 7Doctorado en Medicina de la Conservación, Facultad Ciencias de la Vida, Universidad Andres Bello, República 440, Santiago 8370134, Chile; 8Centro de Investigación para la Sustentabilidad, Facultad Ciencias de la Vida, Universidad Andres Bello, República 440, Santiago 8370134, Chile; 9MIVEGEC, IRD, CNRS, Université de Montpellier, 34090 Montpellier, France; 10GEMA, Center for Genomics, Ecology & Environment, Universidad Mayor, Camino la Pirámide 5750, Huechuraba, Santiago 8580000, Chile; 11Institute of Environment, Florida International University, University Park, Miami, FL 33199, USA

**Keywords:** South America, wildlife species, endocrine disruptors, trace elements, organic compounds, metals, ecotoxicology

## Abstract

Human pressure due to industrial and agricultural development has resulted in a biodiversity crisis. Environmental pollution is one of its drivers, including contamination of wildlife by chemicals emitted into the air, soil, and water. Chemicals released into the environment, even at low concentrations, may pose a negative effect on organisms. These chemicals might modify the synthesis, metabolism, and mode of action of hormones. This can lead to failures in reproduction, growth, and development of organisms potentially impacting their fitness. In this review, we focused on assessing the current knowledge on concentrations and possible effects of endocrine disruptor chemicals (metals, persistent organic pollutants, and others) in studies performed in South America, with findings at reproductive and thyroid levels. Our literature search revealed that most studies have focused on measuring the concentrations of compounds that act as endocrine disruptors in animals at the systemic level. However, few studies have evaluated the effects at a reproductive level, while information at thyroid disorders is scarce. Most studies have been conducted in fish by researchers from Brazil, Argentina, Chile, and Colombia. Comparison of results across studies is difficult due to the lack of standardization of units in the reported data. Future studies should prioritize research on emergent contaminants, evaluate effects on native species and the use of current available methods such as the OMICs. Additionally, there is a primary focus on organisms related to aquatic environments, and those inhabiting terrestrial environments are scarce or nonexistent. Finally, we highlight a lack of funding at a national level in the reviewed topic that may influence the observed low scientific productivity in several countries, which is often negatively associated with their percentage of protected areas.

## 1. Introduction

Chemical pollutants have deleterious effects on biodiversity, but several effects are not well-known by society and relevant stakeholders (e.g., policy makers, non-governmental organizations). This phenomenon may be due to the sublethal and chronic effects of many types of chemical pollution, including those denominated endocrine disrupting chemicals (EDCs), which may cause effects through infinitesimally low levels of exposure [1] and with impacts determined many years after the first contamination event including trans-generational effects [2]. South America is a region with increasing population size and urbanized area [3]. This constant growth in population results in an increasing deforestation, agricultural and industrial expansion, along with waste emission [4,5,6,7]. In consequence, there is an increasing use and release of chemicals with unknown impacts on natural ecosystems [8,9,10,11,12]. Therefore, there is a need to quantify and monitor the extent and consequences of this contamination.

Loss of biodiversity has been linked to an increase in environmental pollution [13,14,15], where chemicals emitted by industrial processes, pesticide use, mining and waste discharge are of main concern [16]. Since the last century, over 80,000 chemical compounds have been synthesized, and intentionally or unintentionally released to the environment [17]. Thus, wildlife and humans are exposed to these chemicals through ingestion, dermal contact, respiration, and maternal exposure [18,19,20,21,22,23]. Recent research has questioned the capacity of protected areas to safeguard biodiversity from the effects of chemical pollution [9,24,25], recognizing the need to monitor and control the potential negative effects of these chemicals in wild animals and people. Therefore, we need to quantify and monitor the extent and consequences of this contamination in natural ecosystems. According to The Endocrine Disruption Exchange (TEDX; http://endocrinedisruption.org accessed on 01 January 2021), about 1000 chemicals are recorded as EDCs (e.g., plastics, personal care products, pesticides, metals, biogenic and industrial chemicals) [26]. Most of these chemicals are released as a combination of EDCs into the environment every day and can negatively affect and disrupt the endocrine system of wildlife species [27,28,29,30]. In addition, new chemicals are manufactured and enter the market, without quantifying their possible effects on wildlife and/or humans [20,31,32]. EDCs category includes persistent organic compounds (e.g., organochlorine pesticides, polychlorinated biphenyls (PCBs), polybrominated biphenyls (PBBs), brominated flame retardants (PBDEs), dioxins), detergents, plasticizers, and plastic additives (e.g., nonylphenol), bisphenol A (BPA), diethylstilbestrol, persistent halogenated hydrocarbons (PHAs) and tributyltin (TBT), plasticizers and plastic additives [33,34,35,36], and Perfluoro alkyls (PFOS and PFOA). In addition, some metals (Cd, Pb, Hg, As) are considered EDCs due to their adverse effect on health of different species including humans [31,34,36,37,38]. One mode of action (MoA) of EDCs is to interact with the hormonal system by binding with endocrine receptors, which either block, magnify or inactivate the subsequent events of hormone action in an organism [19,39,40,41]. These alterations represent a series of “false signals” that can modulate the normal endocrine function at low dose exposure [42,43,44,45,46]. The consequences of endocrine disruption include alteration on reproduction (e.g., low fertility rates, quality of the sperm, imposex), development (e.g., malformations, growth, body mass, immune system impairment) behavior (e.g., communication skills, mating, feeding times, predator-prey dynamics) of the exposed organisms, but also their offspring [27,28,33,47,48], which highlights potential long-term consequences for the conservation of wild species.

The purpose of this review is to compile the available literature on EDC effects on South American wildlife published from 1985 to 2019, from a country contribution point of view, including (1) studies on the concentration in animal tissues/organs/acellular structures of EDCs at the individual-level, (2) effects of EDCs on an individual’s development and reproduction, and (3) the use of biomarkers to determine EDCs’ impact on an individual’s development and reproduction. This work provides a necessary update of knowledge on EDCs impact on organisms (both vertebrates and invertebrates) in the region identifying relevant gaps that can be filled with future research.

### 1.1. Endocrine Regulation and Effects of Xenobioc Chemicals

Vertebrates have three major neuroendocrine systems controlling reproductive processes, growth, development, and metabolism: (a) the hypothalamic–pituitary–gonadal (HPG) axis, (b) the hypothalamic–pituitary–thyroid axis (HPT), and (c) the hypothalamic–pituitary–adrenal axis (HPA) [30]. The hypothalamus regulates the endocrine system and initiates the secretion of the hormones for each of the three axes. Cross regulations between the HPT axis and the HPG axis influence reproduction and metabolism [48,49,50]. Additionally, cross regulations between the HPT axis and the HPT axis will influence the development and metabolism [48,49,51].

EDCs also act at different levels of the endocrine axes, including their feedback mechanisms, which may lead to physiological malfunctions. Hormone regulation and production are modified by EDCs (Figure 1) [27]. Since all systems are interconnected, consequences of EDCs on one system may have effects on multiple compartments, leading to failure at the individual level with implications on metabolism, growth, reproduction and/or development [48,49,51,52,53,54,55,56]. However, for this review, only the HPG and HPT axis were evaluated. Since these two axes are the main and most studied in wildlife, it can be understood how an alteration in the action of hormones will affect the metabolic regulation of the processes involved in growth, reproduction, and behavior. This directly affects the fitness of the species in the wild.

#### 1.1.1. The Hypothalamic-Pituitary-Gonadal (HPG) Axis

The HPG axis controls the sexual steroids production, preparing the organism for reproduction. The hypothalamus secretes the gonadotropin-releasing hormone (GnRH) and the gonadotropin inhibitory hormone (GnIH), which regulates the secretion of gonadotropins: the luteinizing hormone (LH) and the follicle-stimulating hormone (FSH) in the pituitary [27,51,57]. In turn, the LH and the FSH induce the secretion of steroid hormones (testosterone and estrogens) in the gonads, having a direct effect in the target tissues [51,57]. The steroid hormones control the release of the GnRH and gonadotropins to maintain accurate concentrations of FSH and LH via negative feedback [57,58]. The EDCs compounds influence the secretion of GnRH, LH, FSH and have an influence on the enzymes responsible for the conversion of testosterone to dihydrotestosterone or 11-keto-testosterone (or androgen similes, depending on the organism) in males for testosterone to estrogen in females (Figure 1) [58,59,60]. EDCs that have an estrogenic or antiandrogenic activity will impact the testis, leading to their abnormal development and a possible feminization in male organisms, while androgenic EDCs will cause masculinization in female organisms by influencing female gonads [58,60,61,62]. Antiestrogenic compounds in male and female gonads have a harmful effect on gonadal development [58,63].

During embryonic development, EDCs interfere in the reproductive neuroendocrine axis provoking permanent effects on physiology and behavior in adults with failure in growth, development, and reproduction [21,48,64]. Reproductive dysfunctions in wildlife include alteration in fertility and changes in their reproductive anatomy, reducing normal hormone secretion and future generations’ viability [27,28,30]. In the long term, changes in reproductive aspects can lead to the viability of future generations [65]. For example, EDCs caused changes in testosterone and estrogen levels that led to low fertility and alterations in reproductive behavior and ultimately caused the population decline during several decades for the alligator (*Alligator mississippiensis*) in Lake Apopka, Florida (United States) [28,66,67,68]. Similarly, EDCs on sewage effluents developed intersex (i.e., eggs within the testis) on fish [69], and organotin compounds like tributyltin (TBT) are related to imposex (female masculinization) in gastropods worldwide, leading to reproductive failure and population decline [70,71].

#### 1.1.2. The Hypothalamic-Pituitary-Thyroid Axis

The HPT axis regulates the secretion of thyroid hormones, which are essential for metamorphosis in amphibians, and the development and metabolism in all vertebrates [53,59,72,73,74]. The hypothalamus secretes the thyrotropin-releasing hormone (TRH), which induces the secretion of the thyroid-stimulating hormone (TSH) in the pituitary [51,55,57]. TSH stimulates the secretion of the thyroid hormones: thyroxine (T4) and triiodothyronine (T3); and enhances iodine accumulation, which is necessary for the biosynthesis of thyroid hormones (Kloas and Lutz, 2006). TSH also induces the enzyme thyroid peroxidase (TPO), also necessary for thyroid hormone production (Kloas and Lutz, 2006). The thyroid hormones control via negative feedback the release of the TRH and the TSH [57,58]. In amphibians, the hypothalamus secretes the corticotropin-releasing hormone (CRH) instead of the TRH [55]. Studies in different species of terrestrial and aquatic taxonomic groups such as frogs (e.g., *Rana pipiens*), fish (e.g., *Clarias gariepinus, Danio rerio*) and rats (e.g., *Rattus norvegicus*) revealed that thyroid hormones affect the synthesis and action of steroid hormones, having an impact on gonadal differentiation, hormone levels and reproduction [53,75,76,77,78,79,80]. At the HPT axis, EDCs will have a negative effect at different stages of the synthesis, secretion and metabolism of the thyroid hormones influencing eventually their serum concentration and thyroid function [58,81,82]. Thyroid hormones are responsible for the metamorphosis of amphibians and a disruption on HPT axis could lead to an accelerated or incomplete metamorphosis, affecting their development and reproduction and leading to population decline (Figure 1) [55,74,83,84,85]. In reptiles, disruption of thyroid hormones has consequences in gene expression, thermoregulation, reproduction, and metabolism [38,86]. In birds, a study in crows (*Corvus macrorhynchos*) revealed histopathological thyroid gland changes related with environmental chemicals in an urban area [87].

### 1.2. The Use of Biomarkers of Endocrine Disruptors

Biomarkers are used to assess whether an organism has been exposed to a toxic compound and to detect possible effects in tissues and susceptible individuals in different ecosystems [88]. Biomarkers are quantifiable changes in organisms at morphological (e.g., morphological, and histological thyroid gland observation in Xenopus laevis (African clawed frog)), physiological (e.g., body size, weight and bridal pad in Bufo bufo) or biochemical levels (e.g., induction 7-ethoxyresorufin O-deethylase (EROD) in liver of Perca fluviatilis) [89,90,91]. Biomarkers are subdivided into three types [88,92,93]: (I) Biomarkers of exposure, which indicate a direct exposure of an organism to a pollutant (measure of a contaminant or its metabolites in biological tissues of an organism like organochlorine concentrations in dolphin tissue [94]). (II) Biomarkers of effect, which are the biological responses of the organism related to the exposure to a contaminant, where physiological or biochemical changes are detected (e.g., histological alteration in fish exposed to methyl mercury [95]). (III) Biomarkers of susceptibility, which identify susceptible individuals in a population exposed to a specific pollutant (e.g., gene polymorphism due to exposure of mercury [96]).

In this review, we focused on the EDC effects in the two main axes (HPG axis and HPT axis) in wild animals of South America, using the biomarkers described above. The results were reported considering the country and the species studied, as well as different EDCs identified.

## 2. Materials and Methods

Data were obtained from publications related to EDCs from 1985 to 2019, which were quantified in different organs of animals excluding those without having considered any physiological changes in the species in South America. Subsequently, a search for publications related to the presence of endocrine disruptors in tissue/organs or animal parts associated with alterations in reproduction, growth or development was made. Publications under experimental studies were incorporated. Additionally, publications with invasive species have been included as a surrogate to native species, especially due to a lack of knowledge of the biology/physiology of native wild species. The papers published in various academic databases (ISI Web of Knowledge, Google Scholar, Scopus) were searched using the keywords like “Persistent Organic Pollutants” or “Endocrine Disruptors” or “Heavy Metal”. In addition, keywords such as “Fauna” were combined with the names of the countries of South America (e.g., “Persistent Organic Pollutants” or “Endocrine Disruptors’’ and “Fauna” and “Chile “). We excluded from our search papers that report levels and/or concentrations as routine monitoring without any physiological changes related to endocrine alterations. Later, we commented on our results as reported by each country, revealing regional differences.

## 3. Results and Discussion

A total of 606 scientific articles reported the concentrations of EDCs in South America between 1985 and 2019. The 605 publications are distributed in thirteen countries: Argentina (88) Bolivia (7), Brazil (325), Chile (72), Colombia (47), Ecuador (6), French Guiana (11), Paraguay (1), Peru (14), Suriname (2), Trinidad and Tobago (3) Uruguay (5) and Venezuela (25) of the 14 countries surveyed (Table 1). Among the reviewed publications, 72.3% (438) assessed the concentration of metals in tissues of animals, 20.8% (126) were based on persistent organic compounds, 2.3% (14) analyzed concentrations of metals and persistent organic compounds together, 0.6% (4) articles evaluated persistent organic compounds and other compounds and 4% (24) publications on several other compounds (Table 1).

The number of publications increased exponentially over the last 30 years, being Brazil the principal contributor (Figure 2). Brazil, Argentina, Chile and Colombia contributed in the last ten years the most of the publications related to the measurement of concentrations of metals and persistent organic pollutants (POPs) in species tissues (Figure 3). Most publications focused on fish 41.1% (249) followed by more than one class of organism 12.2% (74), mammals 12% (73), birds 10.6% (64), bivalve 10.6% (64), crustacean 4.5% (27), reptile 4.3% (26), gastropods 3.1% (19), insects 0.6% (4), and amphibians 0.3% (2) (Figure 4). The larger proportion of studies focusing on fishes could be linked to their economic importance as well as the possible risk for human health, due to exposure to this type of contaminant via ingestion. A total of 8.9% (54) publications studied showed the presence and concentration of metals in more than one animal class, revealing the biomagnification of the compounds through the food web (Figure 4). Most studies in mammals were focused on marine species (Appendix A). Studies conducted in mammals include two publications related to measurements of mercury in otter in Brazil and Peru [97], and three related to metals concentrations in tissues from bats, jaguars, and wild canids in Brazil [98,99,100,101,102]. Two papers focused on measurements of metals in jaguar and wild mice from Colombia [103,104], and one paper that determined organochlorine pesticide concentration in the tissue of a guinea pig in Argentina [105].

Most papers focused on concentrations of metals, POPs and other compounds, which that cannot be compared between countries, since authors reported the concentrations of contaminants in different types of tissues (fat and muscle, hepatic, gonadic), egg parts (yolk, albumen, shell, and whole egg), acellular structures (mainly proteinic, feather, carapace), blood, and the whole animal in the case of invertebrates and different weight references (wet weight, fresh weight, dry weight, lipid weight), which makes it difficult to compare concentrations and effects [106,107,108,109,110]. In some cases, studies reported that the levels of pollutants were too low to cause a population decline or concentrations found in tissues of organisms did not exceed the limits set by environmental authorities. None of the studies described a relationship of concentrations of POPs, metals, and other compounds with an effect on the HPG axis nor the HPT axis.

On the other hand, twelve studies assessed the concentrations of organotin compounds found in tissues and the incidence of imposex in gastropods (Appendix A). Additionally, 30 studies about imposex in gastropods were related to their possible exposure to organotin compounds present in the environment (Appendix A). Twenty-one (21) studies focused on the effect that EDCs may have on wild populations or an invasive species. Another 47 studies focused on experimentally assessing the impact of EDCs (compounds, polluted sediments, polluted water) on different species at different developmental stages using biomarkers as endpoints. Argentina, Brazil, Chile, Colombia and Venezuela revealed most of the publications and will be explained individually in the following section.

### 3.1. Argentina

Most publications in Argentina 66% (58) focused on metals, followed by POPs 25% (22). A small proportion was related to the measurement of concentrations of organotin compounds in gastropod tissues where imposex was also evaluated [111,112,113]. Fish and mammals (cetaceans and pinnipeds) were the principal taxonomic groups selected to assess the concentrations of EDCs (Appendix A). A study in Argentina revealed imposex in gastropods and butyltins, PAHs, and POPs accumulation in sediments and bivalve muscle [114]. No study focused on measuring metals or POPs in wild reptiles or insects was found for this country.

In five studies of Argentina, imposex in gastropods was evaluated in marine areas, indicating their possible exposure to organotin compounds present in the environment (Appendix A). Imposex was reported in gastropods in the coastline of Argentina and in the edible snail (*Adelomelon ancilla*) in two sites in Golfo Nuevo [115,116]. One study evaluated the concentrations of TBT that ranged from not detected to 1369.58 ng Sng^−1^ in sediments in different study sites of the Argentinean shoreline and their incidence of imposex [117]. Another study was conducted in areas where TBT was previously detected (up to 174.81 ng Sn g^−1^ DW) and shell shape was associated to imposex in gastropods [118]. Shell shape was used to evaluate marine pollution through history in *Buccinanops globulosus* [119] (Appendix A).

Three studies in Argentina reported the possible effects of EDCs on wild fauna using biomarkers. For example, the organochlorine pesticide concentrations were evaluated in tissues of fish-eating birds (ΣHCH range: ND to 3168.41 ngg^−1^ fat, ΣCHL range: ND to 4961.66 ngg^−1^ fat, ΣALD range: 287.07 to 9161.70 ngg^−1^ fat, ΣDDT range: 1068.98 to 6479.84 ngg^−1^ fat) and amphibians (heptachlors: 2.34 ± 0.62 ngg^−1^ wet mass, hexachlorocyclohexanes: 9.76 ± 1.76 ngg^−1^ wet mass) in the Reservoir Florida, along with possible effects in the biota [120,121]. In the birds and amphibians of that study, possible internal and external malformations were evaluated, but no possible relationship with the POPs was found [120,121]. In the introduced fish (*Gambusia affinis*), several biomarkers such as histopathological parameters, vitellogenin expression and copulatory organ morphology revealed alterations in different gradients of water quality in the Suquía River basin [122]. In water samples, alpha-cypermethrin was detected from lower than the detection limit to 23.4 ± 7.70 ng L^−1^, beta-endosulfan from lower than the detection limit to 4.6 ± 1.8 ngL^−1^, chlorpyrifos from lower than the detection limit to 3.3 ± 0.5 ng L^−1^, endosulfan-sulfate from lower than the detection limit to 5.1 ± 2.6 ngL^−1^ and mercury from lower than the detection limit to 0.33 ± 0.02 ng L^−1^ [122].

Twenty-six (26) publications in Argentina were based only on experimental studies at the laboratory level to observe biomarkers or biological alterations to environmental contaminants that act as EDCs in different species at different developmental stages (Table 2). Most of the publications were based on the reptile *Caiman latirostris* and fish (Table 2).

### 3.2. Brazil

Most of the literature regarding EDCs in South America has been published from Brazilian studies, counting 322 publications, reporting different concentrations of EDCs found in tissues of animals in Brazil. The majority, 76% (247), were recorded for metals, 18.2% (59) assessed POPs levels, 2.2% (7) studied metals and POPs together, 0.6% (2) studied POPs with other compounds together and 3% (10) for other compound concentrations. Diuron, as well as chlorinated pesticides and PCBs were quantified and related to immunological and pathological findings in the liver of fish [149]. Organotin compound concentrations were analyzed in cetaceans, fish, gastropods, crustaceans and ascidiacea [150,151,152,153,154,155,156] (Appendix A). Gastropods were used as bioindicators revealing imposex or shell shape differences due organotin compounds present in the environment in fourteen studies, without measurements of these compounds in their tissue (Appendix A).

The possible effect of EDCs in wildlife has been reported in 12 publications of Brazil. This included the response of wild fish (*Astyanax fasciatus*) exposed to discharges from agriculture, industrial and municipal wastewater in Furnas Reserve [157] Biomarkers such as feminization index, intersex rate, reduction in body size, delayed gonadal maturation, increase in proteins of the zona radiata and increased liver-somatic index, were assessed in an exposure gradient of sampling sites of the river basin [158].

Alterations, such as incidence of histopathological changes, expression of metallothionein, vitellogenin and radiata zone protein were related to the concentration of metals in water and fish (*Prochilodus argenteus*) in a polluted river in Brazil [157]. Additionally, water conditions impacted by anthropic activity indicated a higher concentration in plasma E2 levels and hepatic vitellogenin gene expression in males, as well as an absolute and relative fecundity in females [159].

The total estrogen level in the water (<120 ng L^−1^) had an impact on vitellogenin levels, zona radiata (eggshell) proteins, growth factors like insulin (IGF-I and IGF-II) and reproductive parameters in wild male and female fish Astyanax fasciatus exposed to discharges of untreated municipal and industrial sewage [160]. Hermaphroditism in frogs (*Physalaemus cuvieri*) was observed when exposed to waters with concentrations around 0.05 mg L^−1^ of dieldrin [161]. Similarly, high levels of estrone (187.39 ng g^−1^), estriol (34.68 ng g^−1^), diethylstilbestrol (453.69 ng g^−1^), 17α-estradiol (1.36 ng g^−1^), 17α-ethinylestradiol (70.28 ng g^−1^) and 17β-estradiol (52.82 ng g^−1^) in sediments of the Pacoto River (Ceará, Brazil) induced the vitellogenin expression in male fish (*Sphoeroides testudineus*), although the gonads of the fish had a normal structure [162]. Likewise, concentrations of environmental estrogens (estrone (>250 ng L^−1^), estradiol (>200 ng L^−1^), estriol (>200 ng L^−1^), bisphenol-A (>190 ng L^−1^) and nonylphenol (>600 ng L^−1^) in different sites of collection produced follicular atresia, yolk deficient oocytes, over-ripening and decreased vitellogenin in female fish (*Astyanax rivularis*), intersex gonads and vitellogenin induction in males [163]. Fishes (*Astyanax rivularis*) in the Velhas River headwaters with estrogenic compounds in water (estrogenic potential (EEQt) of S1 site was 161.7 ng L^−1^, S2 site: 667 ng L^−1^ and S3: 1300 ng L^−1^) showed alterations in gonad morphology, and changes in germ cell proportion and on the sex steroid levels [164]. Male fish captured from the Iguaçu River exhibited increased levels of vitellogenin, and female fish revealed decreased levels of vitellogenin and estradiol, and immature gonads and degeneration of germ cells [165].

An article demonstrates that Cu (0.035 mg kg^−1^) in sediments had a negative effect on the survival of embryos of the sea turtle *Erehmochelys imbricate* and an increase of Ni (1.711 mg kg^−1^) in adult female blood was responsible with fewer eggs in their nests [166]. Variations of concentration of Zn in sediments (highest value in one site: 115 ± 6.9 mg kg^−1^) in urban stream sediments was correlated with deformities in the mentum of chironomid larvae [167].

In Babitonga Bay (Santa Catarina State, Southern Brazil), organotin concentration in tissues (<LOQ to 418.5 ng g^−1^ dw of Sn) was correlated with imposex incidence and total testosterone/total estradiol ratio imbalance the muricid *Stramonita haemastoma* [154]. An article related the butyltin (BT) contamination (383.7 to 7172.9 ng g^−1^ of Sn) that was previously described on the Espírito Santo coast with the incidence of imposex in *L. nassa* and *S. brasiliensis* gastropods [168] (Appendix A). No evidence of imposex was found in the gastropod Stramonita rustica populations of two tropical estuaries in relation to BT concentrations (<LOQ to 542 ng g^−1^ dw of Sn) in sediments [156] (Appendix A).

Eight (8) different studies in Brazil were based only on controlled or semi-controlled exposures to EDCs, to evaluate the response of biomarkers to environmental contaminants (Table 3).

### 3.3. Chile

Most EDCs related published articles in Chile 47.2% (34), reported metal concentrations, 45.8% (33) assessed POPs levels, 4.2% (3) studied metals and POPs together and 2.8% (2) other compounds (Appendix A). Most of the documented research in Chile has been focused on the occurrence of possible EDCs in biota, with a minor approach to effects.

Imposex was found in gastropods that were potentially exposed to organotin compounds in three studies [177,178,179]. The mean value for TBT in sediment samples in different sites ranged from 0.48 ± 0.21 ng g^−1^ to 37.1 ± 26.6 ng g^−1^ and in samples of the biota the values ranged from 0.8 ± 0.3 1 ng g^−1^ to 2.74 ± 0.43 ng g^−1^ [179]. Two studies reported that imposex could be caused by butyltin in sediments and biota, despite a global ban of this component [180,181]. In one study, high TBT concentrations were found in sediments (122.3 ng g^−1^ of Sn) and gastropods tissue (59.7 ng g^−1^ of Sn), while in another study site, TBT concentrations ranged from 7.4–15.8 ng g^−1^ of Sn in biota [181]. The second study revealed TBT levels above of 90 ng Sn g^−1^ in gastropod tissues and 300 ng g^−1^ of Sn in sediments of six study sites [181].

Three articles assessed the possible endocrine disruption effect of industrial effluent discharges in wild fish populations. Several biological responses of freshwater wild fish (*Percilia gillissi* and *Trichomycterus areolatus*) exposed to an industrial pulp mill discharge into the Itata River were reported [182]. The results revealed an increase of 17β-estradiol in females and decreased 11 keto-testosterone in male *Percilia gillissi* and an increase in female gonadal size and an increased hepatic 7-ethoxyresorufin O-deethylase (EROD) activity. Additionally, alterations in fish sizes of both species related to the discharges at different periods of time were detected, which could be linked to the reproductive alterations observed [182]. In the saltwater flatfish (*Paralichthys adspersus*), a decrease in the gonad somatic index was shown, along with changes in male gonadal development, and an increase of plasma vitellogenin and liver somatic index at the seacoast of Itata [183]. Previous studies of the area revealed, among other compounds, a high presence of pentachlorophenol (0.35 ng g^−1^), organic halogens compounds (171.21 mg kg^−1^), total hydrocarbons (3 µg g^−1^) in sediments and aluminum (0.41 ± 0.45 to 3.19 ± 2.41 µg L^−1^), total chromium (0.38 ± 0.26 to 3.70 ± 0.85 µg L^−1^), and copper (0.30 ± 0.48 to 4.66 ± 2.24 µg L^−1^) in the water column along the coastline of Chile [184,185]. Another study in two wild fish species (*Trichomycterus areolatus* and *Percilia iwini*) exposed to paper mill and pulp effluents in the Biobio River revealed an increase in gonadosomatic index and increased hepatic 7-ethoxyresorufin O-deethylase (EROD) related to the estrogenic compound found in the river sediments [186].

One article related the histological changes in male gonads of the invasive amphibian species *Xenopus laevis* to dioxin-like and estrogenic activity in sediments [187]. Bio-TCDD-EQ in sediments from different study sites ranged from 0.003 to 0.69 ng g^−1^ SEQ and Bio-E2 EQ-polar ranged from 0.06 to 5.19 ng g^−1^ SEQ. Additionally, vitellogenin induction and low testosterone concentration were evident in male *Xenopus laevis* from different study sites, indicating their exposure to endocrine disruptors [187].

Six experimental studies at laboratory and semi-controlled scale assess biomarkers or biological alterations in different species due to exposure to EDCs (Table 4).

### 3.4. Colombia

Forty-seven publications from Colombia reported concentrations of EDCs found in the tissues of animals. Of these, 76.6% (36) were recorded for metals, 14.9% (7) articles reported POPs, 4.3% (2) were recorded for POPs and metals, 2.1% (1) assessed POPs and other compounds and 2.1% (1) were related to other compounds (Appendix A). The articles for other compounds evaluated organochlorine and organophosphates pesticides in fish at the Bogotá River in Suesca and the presence of perfluorinated compounds in fish (*Mugil incilis*) and in tissues of pelicans (*Pelecanus occidentalis*) [194,195]. One study showed imposex in gastropods, *Stramonita haemastoma*, with possible exposure to organotin compounds [196] (Appendix A).

Two articles focused on the possible effects of EDCs in wildlife populations. The study evidenced an increase in calcium (9.91 ± 0.65 ng g^−1^) and mercury (19.86 ± 1.88 ng g^−1^ r) concentrations in eggshells, reduced eggshell thickness, lesser weight and length of eggs from egrets (*Egretta thula*) when compared to more pristine egret’s nesting areas [197]. A study conducted in conjunction in Colombia and Nicaragua showed disturbances in oysters (*Crassostrea*) reproduction (gamete development, alterations in sex ratio) related to pollutant exposure in Isla Brujas, Taganga and Isla Barú in Colombia [198].

A laboratory conducted article revealed that cadmium exposure at environmentally relevant concentrations (0.0025 ppm) caused damage of sperm quality and changes in the initial stages of development of in the fish *Prochilodus magdalenae* [199]. Additionally, a laboratory conducted study with tropical cup oysters (*Saccostrea sp*.) revealed an anti-estrogenic effect of Cd at high concentrations (1000 µg L^−1^), where vitellogenin was lower compared to the control group [200]. An experimental study of water samples with potentially toxic xenobiotic substances revealed in *Chironomus columbiensis* deformities in the mentum and wing [201].

### 3.5. Venezuela

Twenty-five studies (25) showed concentrations of EDCs in fish, bivalves, birds, crustaceans and gastropods. Of these, 88% (22) focused on metal concentrations, 8% (2) analyzed POPs in different tissues and 4% (1) of the articles reported other compounds (Appendix A).

One laboratory experiment exposed three species (*Pseudoplatystoma fasciatum, Piaractus brachypomus* and *Colossoma macropomum*) of male fishes to estradiol to characterize vitellogenin through a proteomic study [162]. The study points out that peaks of vitellogenin spectra for *C. macropomum* (*m*/*z*: 1481.7, 1537.9, 1649.9), *P. brachypomus* (*m*/*z*: 1546.8, 1573.8, 1621.9) and *P. fasciatum* (*m*/*z*: 1642.9, 1665.9, 1706.0) were significant [202].

One article revealed levels of butyltin compounds (<LOQ to 53.6 ng g^−1^ of Sn) in gastropod *Plicopurpura patula* visceral tissue in different study sites and the incidence of imposex [203]. One article related imposex in *Voluta musica* with the presence of TBT (3.9 ± 3.4 ng g^−1^ of Sn) and Cu (21.9 ppm) in the sediments, and another assessed imposex in gastropods without evaluating TBT compounds in sediments or water [204,205] (Appendix A).

### 3.6. Other Countries

For other South American countries including Bolivia, Ecuador, French Guyana, Paraguay, Perú, Surinam, Trinidad & Tobago and Uruguay, a total of 39 articles on EDCs were found. Most studies focused on metals 88.7% (39), 6.9% (3) articles for POPs in mammals, 2.2% (1) article for POPs and metals together, and 2.2% (1) on concentrations of tributyltin compounds in tissues and imposex in *Thais ípicamen* [150] (Appendix A). Imposex incidence was evaluated in marine snail *Xanthochorus buxea, Thaisella chocolate, Xanthochorus buxeus* and *Stramonita chocolate* in Peru, as well as in muricid species such as *Thais biserialis, T. brevidentata, T. kiosquiformis*, *T. melones, Plicopurpura patula* and *Plicopurpura columellaris* in Ecuador [206,207,208,209,210] (Appendix A). Tributyltin (TBT), dibutylin (DBT), and monobutylin (MBT) were determined in surface sediments in six coastal areas of Ecuador, and the values ranged between 12.7–99.5 ng g^−1^ dw for TBT 1.8–54.4 ng g^−1^ dw for DBT, and 44–340 ng g^−1^ dw for MBT [206].

In Uruguay, three experimental studies correlated biomarker responses with exposures to polluted sediments or water. The first study found that juvenile fish, *Pimephales promelas*, exposed to water from domestic discharges and pulp mill had no alteration in their testicular structure [211]. The second study showed that juvenile carps (*Cyprinus carpio)* exposed to sediments from urban and industrial effluent discharges along the Uruguay River exhibited delayed testicular maturation, reduced primary spermatozoa, and increased serum vitellogenin [212]. One article related the incidence of masculinized females of the fish *Cnesterodon decemmaculatus* in different sampling sites where urban-industrial and agricultural activities were evident in the Arroyo Colorado basin [213].

## 4. Conclusions and Recommendations

Aquatic wildlife such as fish, bivalves, crustaceans, and marine mammals are the most studied organisms in South America regarding the effects of endocrine disruptor chemicals. The 73% of publications focused on measuring the concentration of metals in different animal tissues, 47% corresponding to fish. Due to differences in the reported units of contaminant concentrations, and types of animal tissue studied in each country, results are difficult to compare across studies and countries of South America. 

Our review shows that even though South America harbors the greatest biological diversity on the planet [214,215,216,217,218,219], evaluations of the EDCs exposure and/or effects on many taxonomic groups such as insects, amphibians, reptiles, birds, and terrestrial mammals are scarce. For example, Colombia is one of the most megadiverse countries and the biggest mercury polluter per capita in the world due to mining activities, where mercury releases to the environment can go up to 150 tons year^−1^ [214,220]. Further, Colombia is the country with more reported chemical pollution cases inside protected areas [25]. Therefore, threats faced by Colombian wildlife from environmental pollution are not fully elucidated, and only very few taxonomic groups (e.g., fish) have been evaluated. The lack of biodiversity investigations related to environmental pollution in Colombia may also be linked to administrative challenges to conduct biodiversity research in this country [221]. 

EDCs are widely distributed in the environment, having negative effects on species of different taxonomic groups, which may affect their population persistence. Our review revealed that, although various compounds that act as endocrine disruptors in tissues of different species of wildlife of South America have been quantified, only some of them (e.g., Brazil and Chile) related their effects at the reproductive endpoint level. Despite the evidence of low concentrations of metals in different fish tissues it has not been determined whether they could be generating or not an adverse effect on the fish reproductive, thyroid, or adrenal health. If fish health is involved, this can also lead to a decrease in their populations and eventually affect human wellbeing.

Studies that assessed EDCs in species at a reproductive level in South America are scarce, and the majority have focused on fish species. Although effects on reproductive health have been assessed, less attention has been given to the endocrine disrupting effects on the HPT axis in wildlife species in South America. Brazil and Chile had publications related to the effects that could have contaminants on wild fish and insect populations [157,167,182].

Among the lessons learned from the present work, we recommend that: (a) the measurements of the levels of EDCs on organisms is an urgent need and should be standardized to allow meaningful comparisons across studies and with other pollutants, (b) further investigations are required on population-level effects of neglected aquatic or terrestrial species in different ecosystems in South American countries to generate crucial information for biodiversity protection, (c) future studies should prioritize research on emerging contaminants (e.g., perchlorate, thiocyanate, nitrates); developing methods to unravel the effects on native species and the use of current available powerful methods such as the OMICs (genomics, transcriptomics, proteomics and metabolomics), and (d) a substantial increase of funding is needed to support research in countries harboring high levels of biodiversity [222].

Finally, the present review identified critical gaps in South America on determining the effects of endocrine disruptors in different ecosystems and wildlife species. Overall, an urgent need for research is necessary to evaluate the impact of mining activities on mammals and several taxonomic groups exposed to pesticides in aquatic and terrestrial habitats.

## Figures and Tables

**Figure 1 toxics-10-00735-f001:**
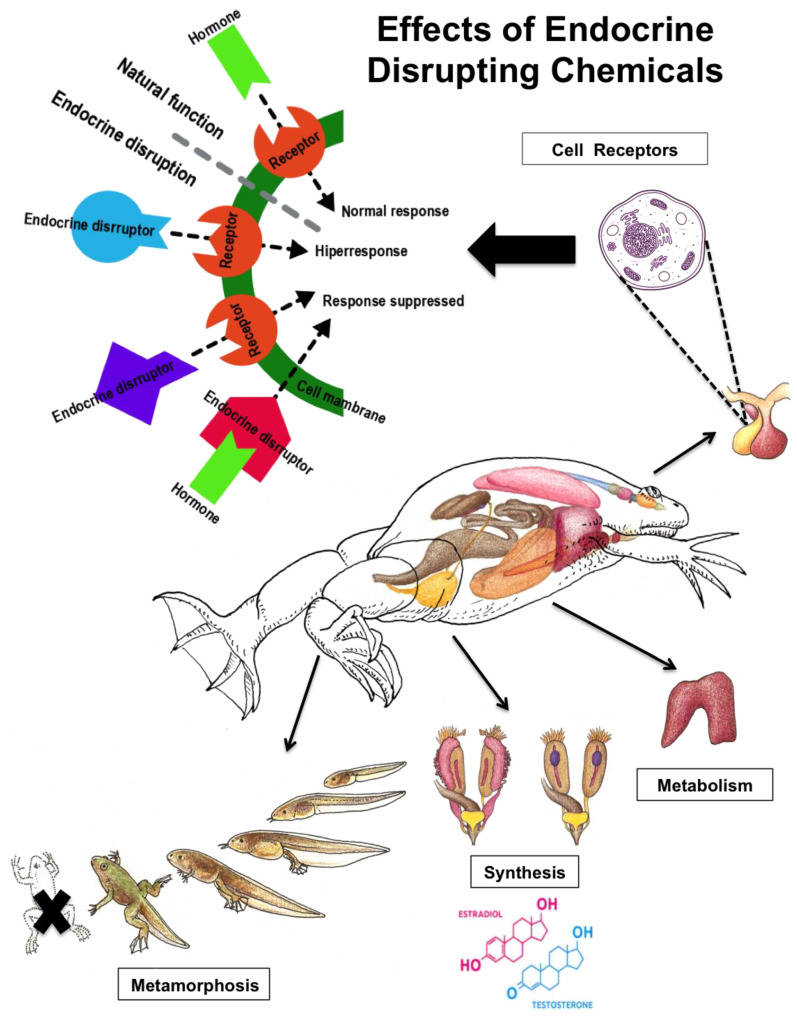
Effects of endocrine disrupting chemicals for vertebrates. Own production figure.

**Figure 2 toxics-10-00735-f002:**
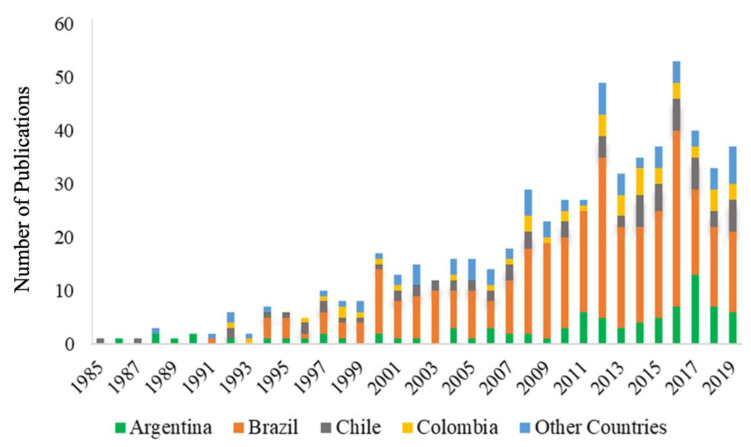
Number of publications by country of concentrations of endocrine disruptors evaluated in tissue.

**Figure 3 toxics-10-00735-f003:**
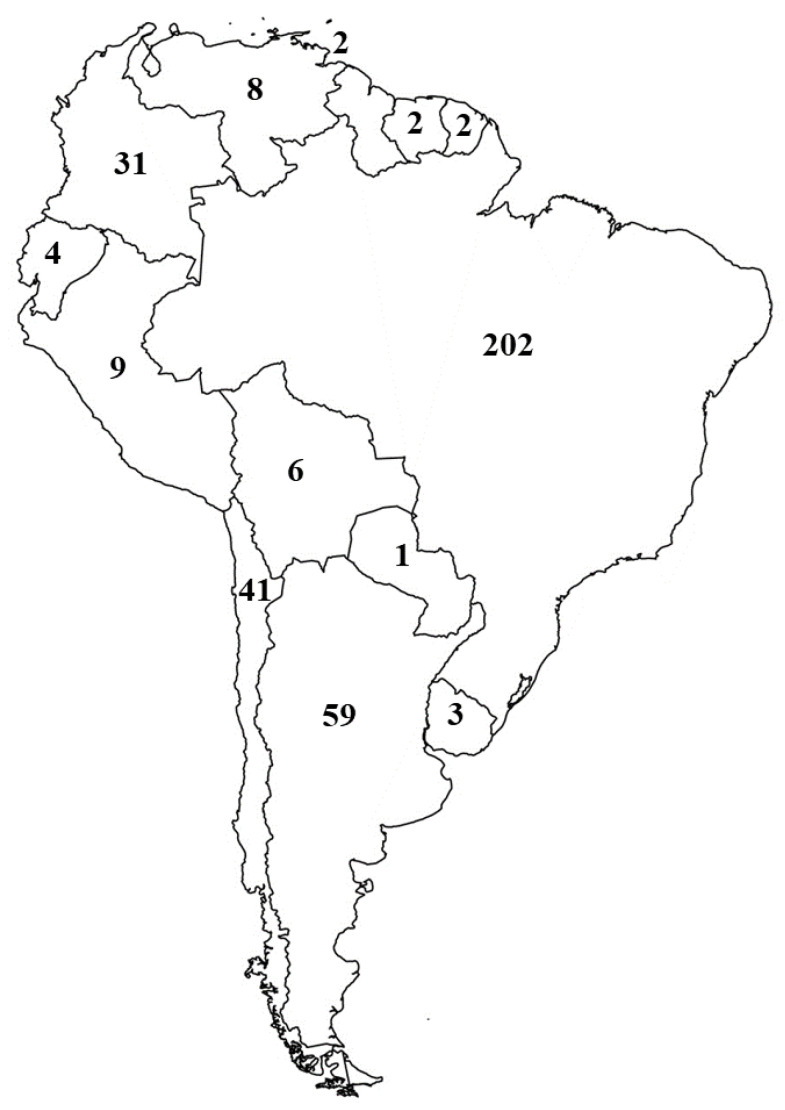
Number of papers in the last ten years in South America.

**Figure 4 toxics-10-00735-f004:**
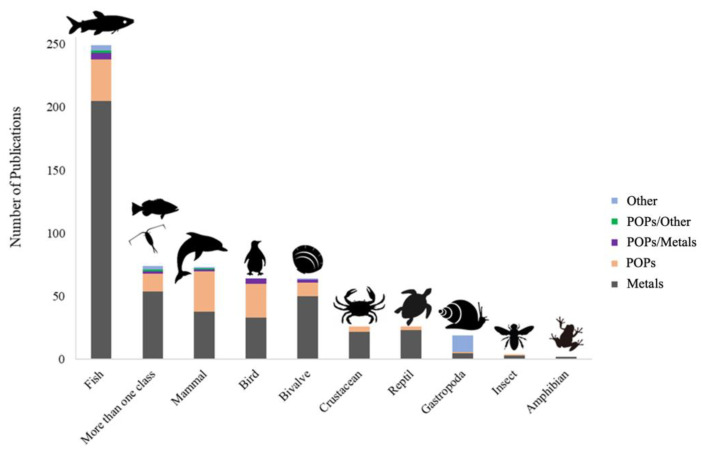
Publications developed by animal group where concentrations of compounds that act as endocrine disruptors were determined. Oligochaeta, Ascidiacea, Chondrichthyes and Polichaeta had one publication each and were excluded from the figure. M (Metals), POPs (Persistent Organic Pollutants).

**Table 1 toxics-10-00735-t001:** Number of publications by country of concentrations of trace metals (TM), persistent organic compounds (POPs) and others. N (%): numbers express the total publications and in parentheses the percentage with respect to the total.

Country	M N(%)	POPs N(%)	POPs/M N(%)	POPs/Other N(%)	Others N(%)	Total N
Argentina	58 (66)	22 (25)	1 (1.1)	1 (1.1)	6 (6.8)	88
Bolivia	7 (100)	0	0	0	0	7
Brazil	247 (76)	59 (18.2)	7 (2.2)	2 (0.6)	10 (3)	325
Chile	34 (47.2)	33 (45.8)	3 (4.2)	0	2 (2.8)	72
Colombia	36 (76.6)	7 (14.9)	2 (4.3)	1 (2.1)	1 (2.1)	47
Ecuador	4 (66.7)	2 (33.3)	0	0	0	6
French Guyana	9 (90)	0	1 (10)	0	0	10
Paraguay	1 (100)	0	0	0	0	1
Peru	11 (78.7)	1 (7.0)	0	0	2 (14.2)	14
Surinam	3 (100)	0	0	0	0	3
Trinidad and Tobago	3 (100)	0	0	0	0	3
Uruguay	3 (60)	0	0	0	2 (40)	5
Venezuela	22 (88)	2 (8)	0	0	1 (4)	25
TOTAL	438 (72.3)	126 (20.8)	14 (2.3)	4 (0.6)	24 (4)	606

**Table 2 toxics-10-00735-t002:** Effects of endocrine disruptors under experimental conditions in Argentina by species, type of contaminant and biological responses.

Species	State of Development	Contaminant	Concentration	Biomarkers or Biological Alterations	Reference
Reptile *Caiman latirostris*	Egg	Endosulfan Atrazine DDT Oxyclordan PCB Bisfenol A 17 β-estradiol	2/20 ppm 0.2/2 ppm Range BDL −153.0 ng g^−1^ lipid 52.0 ± 710.5 ng g g^−1^ lipid Range BDL −34.3 ng g^−1^ lipid 17.8 ± 73.9 ng g^−1^ lipid Range BDL −136.6 ng g^−1^ lipid 23.0 ± 74.0 ng g^−1^ lipid 1.4/140 ppm 0.014/1.4 ppm	Egg weight loss, reduction in hatchling fractional weight, ltered levels of steroid hormones, follicular dynamics, decreased shell porosity and number of eggs per clutch, reduced weight, and size in the young, indirect effect on survival, alteration in gene expression, impaired gonadal histoarchitecture.	[123,124,125,126,127,128,129]
Reptile *Caiman latirostris*	Egg	Endosulfan, Bisphenol A	20 ppm 1.4 ppm	Tortuous seminiferous tubules with empty tubular lumens. BPA: Relative seminiferous tubular area was decreased.	[130]
Fish *Cichlasoma dimerus*	Adult (males, females)	Endosulfan, 17*β*-estradiol Octylphenol 4-tert-octylphenol	0.1, 0.3, 1 μL L^−1^ 10 μg g^−1^ body weight dose 50 μg g^−1^ body weight dose 30, 150 and 300 µgL^−1^	Increased synthesis of vitellogenin and zona pellucida proteins, impaired testicular structure.	[131,132,133]
Fish *Jenynsia multidentate*	Adult (males)	4n–nonylphenol	0.20 and 40 µg L^−1^	Gonadsomatic index decreased, multiple apoptotic bodies in Sertoli cells, loss of testicular cystic structure.	[134]
Fish *Jenynsia multidentate*	Adult (males)	17α-ethinylestradiol	10, 75 and 150 ngL^−1^	Reduction in live and motile spermatozoa, increase in dead and immotile spermatozoa and sperm speed, gonadsomatic index decreased.	[135]
Fish *Jenynsia multidentate*	Adult (males)	17β-estradiol	50, 100, and 250 ngL^−1^	Reproductive behavioral: Sexual activity increased at 50 ng L^−1^ E2., but not at other concentrations. No modification in gonadsomatic index and sperm quality.	[136]
Fish *Odontesthes bonariensis*	Larvae	17α-ethinylestradiol	0.1 and 1 µg g^−1^ food	Altered sex ratio, expression of *cyp19a1a* gen increased, expression of *hsd11b2* decreased.	[137]
Fish *Cichlasoma dimerus*	Larvae	17β-estradiol, 4-*tert*-octylphenol	1 and 10 μgL^−1^ 10 μgL^−1^	High concentrations of E2: feminizing effect directing sex differentiation towards ovarian development. Lower concentration of E2: testis development was inhibited. Exposure, no impairment of male gonad development and functionality.	[138]
Fish *Cichlasoma dimerus*	Adult (male)	4-*tert*-octylphenol	150 and 300 μg L^−1^	High concentration of OP: Impairment of testis architecture. Fishes were transferred to OP-free water after 60 days of exposure: at day 28 testicular functionality was recovered.	[139]
Fish *Cichlasoma dimerus*	Adult (males, females)	Endosulfan	ES-AI: 100 μM	ES by itself did not affect testosterone and estradiol levels. ES with an active ingredient caused steroidogenesis disruption.	[140]
Fish *Odontesthes bonariensis*	Adult (male)	Metals	Cd 0.25 μgL^−1^ Cr 4 μgL^−1^ Cu 22 μgL^−1^ Zn 211 μgL^−1^	Laboratory exposure to environmental concentrations of Cd, Cr, Cu and Zn. Gonads of the fish exposed to all the tested metals suffered structural damages showing shortness of the spermatic lobules, fibrosis, testis ova and the presence of yknotic cells. With Cd: increased expression *gnrh*, Cd and Cr: decrease of *fshb.*	[141]
Crustacea *Zilchiopsis collastinensis*	Adult (females)	Endosulfan	94 ± 6; 192 ± 10 and 360 ± 15 μg endosulfan L^−1^	Changes in volume of oocytes in a certain period without change in the gonadsomatic index.	[142]
Crustacea *Cherax quadricarinatus* (invasive species)	Juvenile	Atrazine	0.1, 0.5 and 2.5 mgL^−1^	Weight gain decreased. At higher atrazine concentration the proportion of females increased gradually.	[143]
Crustacea *Neohelice granulata*	Adult (female)	Atrazine	0.03, 0.3 and 3mgL^−1^	Higher proportion of previtellogenic oocytes, reduction, and delay in the ovarian growth, vitellogenin decreases.	[144]
Crustacea *Eurytemora americana*	Adult (females)	Sewage effluents (4 different water qualities)		Fertility was reduced at bioavailable contaminants from dissolved phase of the sewage effluent.	[145]
Amphibian *Rhinella arenarum*	Adult (males, females)	Cadmium	0.5 and 5 mg kg^−1^	Ovary: nuclear and cytoplasmic pleomorphism, vacuolization of oocytes in the early stages of development. Higher dose: increase in the proportion of atretic oocytes. Testes: seminiferous tubules dilated, disappearance of cysts, leukocyte infiltration. Decreased concentration, viability, and progressive motility of sperm	[146]
Amphibian *Rhinella arenarum*	Larvae	Fludioxonil, Metalaxyl-M	0.25 and 2 mg L^−1^	General underdevelopment, gonadal development and differentiation were impaired.	[147]
Amphibian *Leptodactylus latrans*	Larvae	Glyphosate Roundup ^®^	3–300 mgL^−1^ 0.0007–9.62 mg of acid equivalentsL^−1^	Oral abnormalities and edema. Swimming activity affected.	[148]

**Table 3 toxics-10-00735-t003:** Effects of endocrine disruptors under experimental conditions in Brazil by species, type of contaminant and biological responses.

Species	State of Development	Contaminant	Concentrations	Biomarkers or Alterations	Reference
Mussel *Perna perna*	Adult (males, females)	Coastal area of São Paulo		Reduction in embryonic development, negative impact on the community structure at one study site.	[169]
Fish *Gymnotus carapo*	Adult (males)	Mercury chloride	5–30 μM	Reduction in sperm count and impaired sperm morphology. Direct correlation between the accumulation of Hg and severity of lesions.	[170]
Fish *Oreochromis niloticus* (invasive species)	Adult (females)	Diuron Diuron metabolites	100 ng L^−1^	Diuron metabolites: gonadosomatic indices, percentage of vitellogenic oocytes and E2 plasma levels improved. Diuron and its metabolites: germinative cells reduction.	[171]
Fish *Astyanax bimaculatus*	Adult (females)	Endosulfan	1.15, 2.30, and 5.60 μgL^−1^	Increase in diameters of secondary follicles. Secondary follicles: increased expression of integrin β1 and collagen type IV in cytoplasm of follicular cells.	[172]
Fish *Rhamdia quelen*	Adult (males)	Paracetamol	0.25 and 2.5 µgL^−1^	Reduced testosterone levels. High concentration of paracetamol induces estradiol levels.	[173]
Fish *Odontesthes humensis*	Embryos	Glyphosate-based herbicide	0.36 mg a.e.L^−1^	Reduced eye size and distance between eyes after 96 h of exposure.	[174]
Fish *Rhamdia quelen*	Larvae	Water (polluted with PAHs and toxic metals) of Iguaçu River		Skeleton deformities such as lordosis, scoliosis, and kinks in tails. Cranial abnormalities. Thorax injuries.	[175]
Mammal *Artibeus lituratus*	Adult (males)	Endosulfan	1.05; 0.015 (E1) gL^−1^ 2.1; 0.015 (E2) gL^−1^	Decreased plasma glucose concentration and carcass fatty acids.	[176]

**Table 4 toxics-10-00735-t004:** Effects of endocrine disruptors under experimental conditions in Chile by species, type of contaminant and biological responses.

Species	State of Development	Contaminant	Concentrations	Biomarkers or Alterations	Reference
Mussel *Aulacomya ater*	Adult (males, females)	17β-estradiol	1 and 100 μg L^−1^	Increased vitellogenin and some differences in reproductive parameters.	[188]
Fish *Oncorhynchus mykiss* (invasive species)	Juvenile (males, females)	Laboratory exposures to pulp and paper mill effluents and in situ bioassay downstream of the combined discharge of the same pulp mill	10, 35, 60 and 85% [*v/v*]	Higher concentrations of plasma vitellogenin. Male fish revealed intersex characteristics in all the laboratory assays and in caged fish. Increase in the average gonadosomatic index in exposed fish.	[189]
Fish *Oncorhynchus mykiss* (invasive species)	Juveniles (females)	Sediments of different gradients of contamination from the Biobio river impacted by the pulp mill Caged trout exposure to different pollution gradients in the Biobio River Intraperitoneal injection of effluent of a cellulose plant extract	Sediment from the three sampling areas (PRE, IMP, POST), in a 1:10 *w*/*v* proportion	Increase in vitellogenin and gonadosomatic index, presence of vitellogenic oocytes, inhibition of acetylcholinesterase activity and induction of 7-ethoxyresorufin O-deethylase (EROD).	[190,191,192]
Fish *Percilia irwini*	Adults (male, females)	Laboratory exposures to wastewater treatment plant and pulp and paper mill effluents		Increased VTG-like phosphoproteins and hepatic ethoxyresorufi n-o-deethylase induction levels were detected in effluent-exposed individuals.	[193]

## Data Availability

Data are available in the Appendix A.

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
