# Peer review of "South American National Contributions to Knowledge of the Effects of Endocrine Disrupting Chemicals in Wild Animals: Current and Future Directions"

_toxics, 2022, doi:10.3390/toxics10120735_

Round 1
Reviewer 1 Report
The papers deals with an interesting matter and I think is worth of be published. However I have a main concern with the organization of the results by country. I think it could be more significant and easy to follow if it was organized by contaminants, which effects are independent from the country.
Other points.
Introduction page 2: among plasticizers and plastic additives Phthalates are the most studied in humans. Why they are not mentioned?
Point 1.2 page 5, Biomarkers definition.
There is a misclassification (often found in paperes).
All the cited biomarkers are exposure biomarkers, as we are looking at the consequences of a chemical exposure. Among them we can identify three classes:
dose biomarkers are exposure related to the absorbed dose, and are the measure of a chemical or its metabolite in biological compartments of a living organism.
Effect biomarkers are exposure biomarker related to the biological response to the chemical exposure
Susceptibility biomarkers are exposure biomarkers that indicate which individuals are more susceptible to the exposure to a specific chemical due to an individual characteristic, that can be genetic or acquired, and produces an increase of the effective dose.
Reviewer 2 Report
The paper by Rojas-Hucks reviews the literature concerning the effects of EDCs in wild animals of south America published from 1985 to 2019. The topic of the review is interesting because of the threat posed by EDCs to biodiversity. However, my main criticism concerns the way the review was organised. The authors subdivide the review into paragraphs examining, nation by nation, the published papers; I think that, talking about endocrine disruptors, it would be better to divide the review into paragraphs that examine, from time to time, or the different EDCs, and the different works that concern them, or the different organisms used, and the different papers that concern them. This would certainly make the paper more interesting.
My other comments are the following:
Lines 45-57: this paragraph is difficult to understand, with long sentences and very few full stops. It must be rewritten.
Line 50: “thorough” is “through”?
Lines 129-139: “The hypothalamus secretes the gonadotropin- releasing hormone (GnRH) which induces the secretion of gonadotropins…..”. Hypothalamus also release GnIH (McGuire et al., 2013. Gonadotropin releasing hormone (GnRH) and gonadotropin inhibitory hormone (GnIH) in the songbird hippocampus: Regional and sex differences in adult zebra finches. Peptides, 46, 64-75), this must be specified.
Lines 189-191, and 344. Please put in italics genus and species.
Reviewer 3 Report
The manuscript by Rojas-Hucks et al entitled “Effects Of Endocrine Disrupting Chemicals In Wild Animals Of South America: Current Knowledge And Future Directions” provides an in-depth review of existing literature on this topic. The manuscript is well organized, comprehensive in its treatment of the existing literature and generally well written (there will be some need for language editing, but that is a minor issue).
At 27 pages, the manuscript is quite long and distracts from the true review segment. For example, the first 200 lines of the manuscript are entirely introductory and present lots of nice information about EDCs, endocrine axes, etc. However, that is not the focus of the review and not necessary as there are plenty of manuscripts focused on those subjects that could be cited. I would encourage the authors to dramatically shorten this aspect of the manuscript and instead cite the relevant literature. The review itself is comprehensive and accurately reflects the status of knowledge in this field. The greatest shortcoming of the review is it organization by countries rather than classes of chemicals. The authors argue that cross-country comparisons cannot be done because of differences in methodology for tissue quantification (lines 36-37; 248-251; 495-496 and throughout the manuscript). However, when assessing the implications of measured EDC concentrations, the authors frequently refer to studies from other countries (i.e., Canada, USA, Europe) that would suffer equally from a lack of standardization. As a result of the country-centric approach, the impact of the manuscript is diminished and does not present a snapshot of the impacts of classes of EDCs across South America as the title implies. Focusing less on the number of studies by country and class, organizing the review around the classes of chemicals, their concentrations, and effects ACROSS S. America would dramatically increases the value of this manuscript.
Minor revisions:
Figure 1: The resolution of various components of this figure differ from each other (for example chemical structures vs. larval images), suggesting that they may have come from a variety of sources. Unless they are all original to the authors, these sources need to be acknowledged and their permission to be incorporated need to be confirmed.
Line 217: this should be 606 manuscripts I assume?
Table 1: the legend should indicate what each column of data represents – it took me a minute to figure out that the numbers in parenthesis are % values.
Lines 342-344: here aqueous concentrations of estrogens are reported, which is wonderful and useful – I realize the review is focused on tissue concentrations of EDCs, but if aqueous concentrations are provided here, they could be provided for other systems and then allow for cross-country comparison.
Reviewer 4 Report
General COMMENTS:
The review systematizes the bibliography concerning some aspects of endocrine disruption as an environmental hazard in South America. This is a potentially interesting and timely manuscript, but several clarifications are required in terms of the framework and goals (see comments below).
In my opinion, it would be important to unravel a rationale for the option of addressing the problem in South America. It is not enough to start from just the nationality of the authors. What particularities and aspects in common have the geographic areas under analysis in terms of endocrine disruption risks and target species? There is a particular perception of risk in South America that justifies this review, such as the use of chemicals already banned in other regions, the lack of legislation regulating the use of chemicals or its disrespect? Special areas to protect?
Overall, the article is a promising contribution for Toxics, but are several aspects that need to be improved or rethought (see specific comments, below).
SPECIFIC COMMENTS:
The affiliations 10 and 11 are not associated to any author.
Title – Not only effects were reported. Please improve the title in that direction.
Abstract - “that act as endocrine disruptors in animal tissue” – Endocrine disruption is an effect, tendentially, systemic, and, thus, the reference to “tissue” is less adequate.
Introduction
- “Thus, wildlife and humans are exposed to these chemicals through ingestion, dermal contact, inhalation, and maternal exposure” – The exposure pathways mentioned seems to exclude aquatic animals. This is especially relevant considering the prevalence of bibliography devoted to that animal group.
- “(e.g. organochlorine pesticides, polychlorinated biphenyls (PCBs), polybrominated biphenyls (PBBs), brominated flame retardants (PBDEs), dioxins)” - please revise the use of parentheses, in particular avoiding curly/round brackets within curly/round parentheses.
- “The consequences of endocrine disruption include alteration on reproduction, development and behaviour” - this is too narrow as a view of the range of possible effects.
- “The purpose of this review is to compile the available literature on EDCs effects on South American biodiversity” – Biodiversity encompasses much more than just animals.
- “This work provides a necessary update of knowledge on EDCs impact on organisms (…)” – It must be clearly defined the groups of organisms targeted on the review.
- “without having considered any physiological changes in the species in South America” – I disagree from this option, but if the authors persist on this intention, they should justify it.
- “1.1. Endocrine regulation and effects of endocrine disrupting chemicals” – The repetition of words should be avoided.
- “for this review, only the hypothalamic-pituitary-gonadal and hypothalamic-pituitary-thyroid axis were evaluated” – this choice must be justified, desirably using mechanistic motives.
- The hypothalamus secretes the gonadotropin-releasing hormone (GnRH) which induces the secretion of gonadotropins” – The hypothalamus also secretes gonadotropin-inhibitory hormone (GnIH).
- Fig. 1 – the legend must be more complete.
- “Vertebrates have three major neuroendocrine systems” – “Animals” or “wildlife” is not synonymous of “vertebrates”. Again, the groups of species targeted must be coherently assumed and the diversity of neuro-endocrine processes along animal scale should not be ignored.
- It would be useful to define and use abbreviation for the axes (e.g. HPT for hypothalamic–pituitary–thyroid axis).
- “induction ethoxyresorufin o-dietilasa” – please correct to “induction of 7-ethoxyresorufin O-deethylase”.
Material and Methods
Some of the information here presented was previously mentioned. Please avoid repetitions.
Results
- Table 1 – In the caption it is mentioned “TM” while in table it is used “M”. It must be explained what are the values in brackets.
- I would like to see the following question clarified. If a given paper quantified a metal in an animal's tissues without referring to its eventual endocrine-disrupting effect, was it considered in the data shown? Which metals can be excluded as endocrine disruptors if the whole animal scale is considered?
- “in different types of tissue (fat, liver, muscle, egg, gonads, gills, blood, feather, carapace, whole animal)” – Not only tissues were cited (for instance, liver is an organ, not a tissue; feathers are not a tissue).
- “Figure 4. Publications developed by taxonomic group” – Please revise this statement; “Fish” is not a taxonomic group.
- Tables 2, 3, 4… – Concentrations were expressed in different forms (e.g., ppm, μl /L, μg l-1, μM, a.e./L). This drastically reduces the usefulness of these tables.
Round 2
Reviewer 1 Report
I'm satisfied with the author response to my comments
Author Response
Please find attached our response to the comments

Reviewer 2 Report
The paper in its present form is suitable for publication.
Author Response
Please find attached the response to comments

Reviewer 3 Report
Several reviewers (incl. myself) expressed their concern that the organization of the manuscript by country has little scientific meaning. The authors response was to change the title to reflect the national organization, which does not address the underlying concern.
Author Response
Please find our response attached
